# Hepatic Resection Following Selective Internal Radiation Therapy for Colorectal Cancer Metastases in the FOXFIRE Clinical Trial: Clinical Outcomes and Distribution of Microspheres

**DOI:** 10.3390/cancers11081155

**Published:** 2019-08-12

**Authors:** Helen Winter, Joseph Rassam, Pradeep S. Virdee, Rob Goldin, Priyankaa Pitcheshwar, Klara Weaver, John Primrose, David P. Berry, Harpreet S. Wasan, Ricky A. Sharma

**Affiliations:** 1NIHR Oxford Biomedical Research Centre, CRUK-MRC Institute for Radiation Oncology, Department of Oncology, University of Oxford, Oxford OX3 7DQ, UK; 2Centre of Pathology, Imperial College London, London W2 1NY, UK; 3Centre for Statistics in Medicine, Nuffield Department of Orthopaedics, Rheumatology and Musculoskeletal Sciences (NDORMS), University of Oxford, Oxford OX3 7LD, UK; 4Department of Surgery, Faculty of Medicine, University of Southampton, Southampton SO16 6YD, UK; 5Department of Surgery, University Hospital Southampton NHS Foundation Trust, Tremona Road, Southampton SO16 6YD, UK; 6Oncology Department, Faculty of Medicine, Imperial College London, London W2 1NY, UK; 7NIHR University College London Hospitals Biomedical Research Centre, UCL Cancer Institute, University College London, 72 Huntley Street, London WC1E 6DD, UK

**Keywords:** hepatectomy, trans-arterial radio-embolisation, brachytherapy, interventional oncology

## Abstract

The FOXFIRE (5-Fluorouracil, OXaliplatin and Folinic acid ± Interventional Radio-Embolisation) clinical trial combined systemic chemotherapy (OxMdG: Oxaliplatin, 5-fluorouracil and folic acid) with Selective Internal Radiation Therapy (SIRT or radio-embolisation) using yttrium-90 resin microspheres in the first-line management for liver-dominant metastatic colorectal cancer (CRC). We report clinical outcomes for patients having hepatic resection after this novel combination therapy and an exploratory analysis of histopathology. Multi-Disciplinary Teams deemed all patients inoperable before trial registration and reassessed them during protocol therapy. Proportions were compared using Chi-squared tests and survival using Cox models. FOXFIRE randomised 182 participants to chemotherapy alone and 182 to chemotherapy with SIRT. There was no statistically significant difference in the resection rate between groups: Chemotherapy alone was 18%, (*n* = 33); SIRT combination was 21% (*n* = 38) (*p* = 0.508). There was no statistically significant difference between groups in the rate of liver surgery, nor in survival from time of resection (hazard ratio (HR) = 1.55; 95% confidence interval (CI) = 0.83–2.89). In the subgroup studied for histopathology, microsphere density was highest at the tumour periphery. Patients treated with SIRT plus chemotherapy displayed lower values of viable tumour in comparison to those treated with chemotherapy alone (*p* < 0.05). This study promotes the feasibility of hepatic resection following SIRT. Resin microspheres appear to preferentially distribute at the tumour periphery and may enhance tumour regression.

## 1. Introduction

Colorectal cancer (CRC) is the second leading cause of cancer-related death in the United States and Europe. The liver is the commonest site for metastases from CRC and the leading cause of patient mortality. Of the 60–70% of patients who develop colorectal liver metastases (CRCLM), half of these will have a liver-only disease, but most of these patients will be inoperable, due to number or distribution of metastases [1]. Hepatic resection is offered to patients with operable disease as complete surgical resection remains the sole curative treatment for disease confined to the liver, offering the best chance of survival [2]. 

Preoperative chemotherapy can be used to convert unresectable CRCLM to potentially resectable disease [3]. The use of triplet drug regimens and biological therapies has increased radiological response rates and complete marginal (R0) resections, in previously unresectable disease [4,5]. Despite this progress, surgery is only feasible in up to 20% of patients with CRCLM [6,7]. Selective internal radiotherapy (SIRT) with yttrium-90 (Y-90) microspheres is a liver-directed therapy that could potentially improve resection rates by targeting intrahepatic liver lesions in combination with systemic chemotherapy. SIRT exploits the arterialisation of liver metastases to deliver a high tumour-selective dose of intrahepatic radiation. We previously showed in 20 patients that the combination of oxaliplatin-based chemotherapy with SIRT results in high radiological response rates, and two patients were down-sized to subsequent hepatic resection [8]. However, SIRT-related hepatic toxicity has also been described in patients with CRCLM, so the safety of SIRT in the preoperative setting should be established in order to consider this down-sizing combination therapy [9]. 

The FOXFIRE (5-Fluorouracil, OXaliplatin and Folinic acid ± Interventional Radio-Embolisation) trial tested the hypothesis that combining systemic radio-sensitising chemotherapy (OxMdG: Oxaliplatin, 5-fluorouracil and folic acid) with SIRT (radio-embolisation) using Y90-resin microspheres (SIR-Spheres^®^; Sirtex Medical Limited, North Sydney, Australia) as a first-line treatment for liver-only or liver-dominant metastatic CRC will improve clinical outcomes compared to OxMdG chemotherapy alone. Although the combination treatment failed to improve overall survival, it did improve time to progression in the liver [10]. In this report, we examine whether the combination treatment altered clinical outcomes following liver surgery. We also developed a novel method for assessing the distribution of microspheres in resected tissues from those patients who provided voluntary consent (Figure 1), and we analysed the likelihood of histopathological regression in that subgroup of patients.

## 2. Results

### 2.1. Patient Characteristics

A total of 364 patients were randomised in the FOXFIRE study from 13 November 2009 to 31 October 2014 across 28 centres in the UK: One hundred and eight-two to the chemotherapy alone group, and one hundred and eighty-two to the chemotherapy with SIRT group. There were 67% (*n* = 122) liver-only vs. 33% (*n* = 60) liver-dominant metastatic patients in the chemotherapy alone and 66% (*n* = 121) liver-only vs. 34% (*n* = 61) metastatic patients in the chemotherapy plus SIRT group. The degree of liver involvement was ≤25% in 63% (*n =* 114) vs. >25% in 37% (*n* = 68) patients in the chemotherapy alone and ≤25% in 63% (*n* = 115) vs. >25% in 37% (*n* = 67) patients in the chemotherapy plus SIRT group. Full patient characteristics have been reported previously [10]. 

Of the 364 patients, 71 (20%) went on to hepatic resection following first-line treatment with chemotherapy alone or chemotherapy with SIRT. In the chemotherapy alone group, 45% (*n* = 15) received a biological agent concomitantly with chemotherapy (eight received bevacizumab and seven received cetuximab. In the chemotherapy plus SIRT group, 16% (*n* = 6) received a biological agent concomitantly with chemotherapy (four received bevacizumab and two received cetuximab). Median follow-up over the course of the trial among the 71 patients who underwent hepatic resection was 42.7 months.

Forty-seven (66%) male patients and 24 (34%) female patients proceeded to surgery (*n* = 71). The study flow chart is shown in Figure 2. Twelve patients (17%) were known to have extrahepatic disease. Measuring volume of the tumour relative to the liver, 15 patients (21%) had greater than 25% involvement of the liver by the tumour, and 56 patients (79%) had 25% or less involvement. Demographics are shown in Table 1.

### 2.2. Surgical Details

There were 82 surgical episodes among the 71 resected patients; nine patients had two operations, and one patient had three (Table 2). Liver surgery was performed at 16 sites, with three hospitals performing the majority of all the operations. The liver surgery performed during the FOXFIRE clinical trial was described using the Brisbane 2000 classification [11]. Overall, the proportion of FOXFIRE patients to proceed to surgery in the chemotherapy alone group (18%, *n* = 33) and the SIRT combination group (21%, *n* = 38) was not statistically significantly different (*p* = 0.508).

Of the total surgical episodes, four were open-and-closed and did not proceed to resection. The commonest surgical procedures performed were right hepatectomy (*n* = 22), segmentectomies (*n* = 12) and left hepatectomy (*n* = 9) (Table 2). Of the 15 patients who had >25% liver involvement, four had a two-stage resection. Twenty-six patients had complications reported after hepatic resections, the majority grade I and II complications (Table 3). Four patients had grade III complications requiring surgical or radiological intervention. This included one patient who had a pneumothorax; on resection of part of the diaphragm, microspheres were present. One post-hepatectomy liver failure was documented. Two other patients in the combination treatment arm had reported severe complications: Grade IVa endocarditis five months after SIRT and grade IVb pulmonary embolus, myocardial infarction and sepsis 11.5 months after SIRT. The patient with endocarditis had a prolonged admission of 52 days; the patient with multi-organ problems was in hospital for 15 days. 

### 2.3. Time-to-Event Analyses

The median time from randomisation to hepatic resection in the chemotherapy alone arm was 7.8 months (95% confidence interval CI = 6.5–8.5) and in the chemotherapy + SIRT arm was 8.3 months (95% CI = 7.4–9.9). There was no statistically significant difference between the time-to-resection functions for the two treatment groups (log-rank *p* = 0.810) (Figure 3A). There was no statistically significant difference in the rate of trial participants undergoing a hepatic resection between the treatment groups (HR = 0.94; 95% CI = 0.58–1.53).

In total, 59% (42/71) of participants who had a resection had subsequently died from any cause—52% (17/33) in the chemotherapy alone group, and 66% (25/38) in the chemotherapy + SIRT group. Median time from resection to death in the chemotherapy alone group was 25.2 months (95% CI = 21.0-not estimated) and in the chemotherapy plus SIRT group was 21.9 months (95% CI = 19.0–35.7). There was no statistically significant difference in the survival functions between the treatment groups (log-rank *p* = 0.164). There was no statistically significant difference in overall survival among those resected between treatment groups (HR = 1.55; 95% CI = 0.83–2.89) (Figure 3B).

### 2.4. Histopathology Findings

Only patients who signed the voluntary consent for the use of tissue for research purposes could be analysed in the exploratory analysis of histopathology. This equated to 27 out of 71 resected cases being studied. In patients who were treated with chemotherapy plus SIRT, microspheres were present in all samples. Microspheres were typically clustered around small vessels within the tumour vasculature and associated with macrophages (Appendix A). Using a novel method, we developed, the distribution of microspheres was measured in microsphere/mm^2^ in 0–1 mm, 1–2 mm zones within and outside the tumour margin (Figure 1). The highest density of microspheres was found at the 0–1 mm zones, either side of the tumour border followed by the tumour centre (>2 mm into the tumour). The lowest microsphere density was measured in the non-neoplastic liver tissue away from the tumour (Appendix A). In our analysis, there was a statistically significant difference (*p* = 0.034) in the percentage of viable tumour between patients treated with SIRT (median = 0%, range 0–80%, *n* = 11) and those treated with chemotherapy alone (median = 50%, 0–85%, *n* = 11) (Appendix A).

In the patients treated with chemotherapy plus SIRT, a macrophage, foreign body-type giant cell reaction was observed in relation to the majority of the microspheres, confirmed by CD68 staining (Figure 4). This took the form of individual cells and groups of cells. Inflammation and macrophage infiltration demonstrated a non-significant trend to be higher in the chemotherapy plus SIRT group than the chemotherapy alone group (Appendix A). Smooth muscle actin staining to identify small arteries and CD31 staining to identify endothelial cells confirmed that microspheres were not located in small arteries, and very few were in vascular-lined spaces. The presence of microspheres outside the endothelium suggested possible endothelial damage followed by extravasation of microspheres. The vascular changes of sinusoidal obstruction syndrome (SOS) were seen in all cases, as previously described in relation to chemotherapy. Two SIRT cases displayed evidence of severe vascular changes (Figure 4). No severe vascular change was seen in those treated with chemotherapy alone. Sirius red staining identified areas of fibrosis (Figure 4). Tumour fibrosis was more prevalent in the SIRT plus chemotherapy group than the chemotherapy alone group (median 50% vs 10% (*p* = 0.018)). Both arms had cases of fatty infiltration, and there was no significant difference between the groups (Appendix A). In the seven cases in which microspheres were identified in the gall bladder wall, there was no evidence of pathological changes related to these.

## 3. Discussion

This is the first detailed report of a large series of patients treated with SIRT who have undergone hepatic resection in a prospective clinical trial. The results demonstrate the feasibility and clinical outcomes of patients proceeding to surgery after chemotherapy plus SIRT. Interestingly, in a subgroup of the patients proceeding to surgery, we also show that the distribution of microspheres in resected tissue is consistent with the pathological regression of the metastases being treated.

The aims of this surgical series were to describe the hepatic resection rate, the type of surgery and complications following chemotherapy or the combination of SIRT and chemotherapy. The results from the surgical outcomes and histological evaluation of patients receiving hepatic resection from both arms of the FOXFIRE trial have demonstrated the rate of surgical complications and safety following the addition of SIRT is comparable to chemotherapy alone. The SIRFLOX study, with the same trial design as the FOXFIRE study, also reported no significant difference in conversion rates to surgery in the patients receiving chemotherapy alone and those randomised to receive chemotherapy plus SIRT—13.7% (36 patients) compared with 14.2% (38 patients) (*p* = 0.857) [14]. Although the numbers of resected patients are comparable to this study, the SIRFLOX investigators did not provide detailed information on surgical safety/complications, nor an analysis of histopathology. Reassuringly, the incidence of complications after surgery, including a very low incidence of post-hepatectomy liver failure, confirms the findings of a retrospective series by other investigators suggesting that surgery is safe after chemotherapy plus SIRT for CRCLM [12]. The P4S study retrospectively analysed the safety of hepatic resection or liver transplantation in 100 patients with various primary or secondary liver cancers following treatment with SIRT. It concluded that the mortality, complication and liver failure rates were similar in these patients to rates expected in patients who had not received SIRT prior to resection or transplantation. Similarly, a systematic review of the published literature found a conversion rate to hepatectomy of 13.6% in 120 patients and low surgical mortality [15]. Collectively, these studies are very relevant to multi-disciplinary teams considering SIRT to downstage tumours to subsequent resection.

Chemotherapy-induced changes in a normal liver have been well described previously, including steatohepatitis and sinusoidal obstruction syndrome [16,17,18,19] (Appendix A). In one study of neoadjuvant oxaliplatin-based-chemotherapy in patients with CRCLM, 44 out of 87 patients (51%) had sinusoidal dilatation and haemorrhage, with disruption of the sinusoidal barrier compared to none without prior chemotherapy [20]. Nearly half of these patients developed perisinusoidal and veno-occlusive fibrosis. Vascular complications from chemotherapy alone have been described previously: Sinusoidal vasodilatation and congestion, peliosis, haemorrhagic centrilobular necrosis (HCN) and regenerative nodular hyperplasia (RNH) [19]. 

There are limited published data on histopathological changes in specimens from hepatic resections post-SIRT (Appendix A). The histology from a small number of studies has described features following SIRT, including microsphere distribution, tumour response and changes to background liver. In one paper, significant histological features were described [21]. Macrophages associated with a giant cell reaction and radiation changes were described, and the absence of a cellular inflammatory response supported the theory that SIRT caused direct radiation injury to cancer cells [21]. Our study reported here demonstrates clusters of microspheres associated with aggregations of macrophages. This may have some relevance to the mechanism by which the microspheres act on tumour tissue. Chew describes the significant effect that microspheres have on the immune cells and the tumour microenvironment in hepatocellular carcinoma [22]. Further work is required to decipher the effect that SIRT has on the immune cells in the tumour microenvironment in patients with colorectal liver metastases. 

Following resection, it is known that the microspheres are localised within tumour vasculature and that the non-tumour bearing liver can show evidence of portal triaditis with portal and periportal fibrosis. The largest published cohort to date confirmed resin microspheres within the vascular bed of the tumour and portal fibrosis on the adjacent normal liver [23]. Although previous case reports have described microspheres in the gall bladder, we report microspheres present in the gall bladder in seven patients following SIRT. Fortunately, these patients did not report any significant clinical symptoms from the presence of microspheres in the gall bladder, but our findings suggest that persistent upper abdominal pain in patients who have received SIRT should be considered as possibly related to radiation cholecystitis until an alternative clinical cause for the pain is found.

Due to the ethical requirement for separate voluntary consent to analyse histopathology specimens from surgical resection, there were limited numbers of patients for detailed histological analysis. Despite this limitation, we performed an exploratory comparison of histopathological changes in cases treated with chemotherapy alone, versus those treated with chemotherapy plus SIRT (Appendix A). Interestingly, the SIRT cases had a lower percentage of viable tumour than those treated with chemotherapy alone (*p* = 0.034), and tumour necrosis occurred at a higher rate in cases treated with chemotherapy alone (*p* = 0.054). Additionally, the novel methodology we developed for the zonal analysis of metastases demonstrated that the cases, which displayed the highest percentage of viable tumour within the SIRT group, had the lowest microsphere densities within the cohort (Appendix A). This is preliminary evidence in favour of the hypothesis that microsphere density is inversely related to the percentage of viable tumour post-treatment. The high microsphere density we observed at the tumour periphery is consistent with dosimetric studies published over a decade ago [24].

This report represents the most detailed description of surgical and histopathological findings of patients treated prospectively within a clinical trial of chemotherapy plus SIRT for CRCLM. Despite our finding that off-target delivery of microspheres to the gall bladder is frequent, we conclude that surgery after chemotherapy combined with SIRT is feasible. Our novel histopathological analysis provides evidence in favour of the hypothesis that SIRT resin microspheres preferentially distribute at the tumour periphery and that they can enhance regression of liver metastases. 

## 4. Materials and Methods

The FOXFIRE trial was conducted according to the principles of ISO14155, Good Clinical Practice and the Declaration of Helsinki. Ethical approval was granted by the Berkshire Research Ethics Committee (ethics number 09/H0505/1). The trial registration number was ISRCTN83,867,919.

### 4.1. Trial Participants and Treatments

Patients had to be eligible for systemic chemotherapy as first-line treatment for metastatic CRC. Inclusion criteria included: Histologically confirmed CRC with inoperable liver-only or liver-dominant metastases with or without the primary tumour in situ (“inoperable” agreed at Liver Multidisciplinary Team meeting with liver surgical representation), performance status 0 or 1, age ≥ 18 years, life expectancy ≥ three months. All patients provided written informed consent. Exclusion criteria included: Ascites, cirrhosis, portal hypertension (all determined by a clinical or radiological assessment); thrombosis of the main portal vein. Clinical trial details, including eligibility criteria and treatments, have previously been reported in detail [10,25]. 

Systemic chemotherapy consisted of OxMdG (85 mg/m^2^ oxaliplatin infusion over 2 h, l-folinic acid 175 mg or d, l-folinic acid 350 mg infusion over 2 h and 400 mg/m^2^ bolus 5-FU followed by a 2400 mg/m^2^ continuous 5-FU infusion over 46 h). Each chemotherapy cycle lasted 14 days. Protocol chemotherapy was 12 cycles, with the provision to interrupt it for liver surgery. It was mandated that all patients should be discussed at the Liver Multidisciplinary Team meeting with liver surgical representation to consider liver surgery after six cycles of chemotherapy. The oxaliplatin dose was reduced from 85 mg/m^2^ to 60 mg/m^2^ for three cycles around SIRT, based on the phase I–II data previously published by the investigators [8]. Addition of biologically targeted drugs was permitted as previously described [10,25]. A hepatic arteriogram and a liver-to-lung breakthrough nuclear medicine scan were used to assess patient suitability to receive the SIRT. The patient’s body surface area, percentage tumour involvement, and magnitude of liver-to-lung shunting were used to determine the activity (GBq) per dosing charts [25]. Planned SIRT with Y90-resin microspheres (SIR-Spheres^®^, Sydney, Australia), was with cycle 2 day 3/4, although it could be given with cycle 1 or cycle 3 under the circumstances defined in the protocol.

### 4.2. Histopathology and Immunohistochemistry

In patients who gave voluntary consent to study tissue for research purposes, sections of formalin fixed paraffin embedded tissue (liver in 27 cases, gall bladder in 18 cases) were scanned using the Aperio AT Turbo digital scanner (Leica Microsystems, Milton Keynes, UK) and used for the assessment of: Tumour viability and regression, fibrosis and microsphere distribution. The sections were stained with Haematoxylin and Eosin, and Sirius Red to assess fibrosis. An assessment of vascular pathology and fatty change away from the tumour was also performed. 

The density and distribution of microspheres were determined using zonal analysis using Aperio ImageScope (Leica Microsystems, Milton Keynes, UK). An estimation of the density of microspheres counted manually in Non-Neoplastic Tissue (NNT) was calculated using a mean average of the density in two 3 mm × 3 mm to 5 mm × 5 mm areas of tissue for each case (3 mm × 3 mm regions were used in cases where limited NNT was present). A microsphere density for the tumour centre was also counted manually; the tumour centre was defined as the area 2 mm proximal to the tumour border. Sphere densities within the 1 mm and 2 mm zones of the tumour border and the tumour centre were expressed as a ratio of the sphere density calculated for the NNT (Figure 1).

Immunohistochemical staining for CD68 (PGM1, Dako M087601-2), smooth muscle actin (ab12496, ABCAM EPR5368) and CD31 (ab7817, DAKO M0823) were performed on the Leica Bond III automated immunostaining platform using Leica Bond Polymer Refine detection with DAB chromogen (Leica DS9800; Buffalo Grove, IL 60089, USA). Dewaxing and peroxidase blocking were performed on-board as per kit, according to the manufacturer’s instructions. The antibody was applied for 15 min at room temperature, following on-board heat-induced epitope retrieval (HIER) for 30 min at 99 °C with Leica Epitope Retrieval 1 solution (pH 6, Leica, AR9961).

### 4.3. Statistical Analysis

The hepatic resection rate was defined as the number resected in the randomised group divided by the number randomly assigned to that group. Time-to-event analyses were performed in the subgroup of FOXFIRE patients who underwent hepatic resection. For patients who underwent multiple hepatic resections, the first occurrence was used in the time-to-event analyses. The time from randomisation to resection was defined as the time (months) from randomisation to undergoing hepatic surgery. The time from resection to death was defined as the time (months) from undergoing hepatic surgery to death by any cause. Patients not recorded as dead following surgery were censored at their last known alive date. Surgical complications were graded as previously published [13], and grades included any deviation from normal post-operative course to grade 4-organ failure. A Fisher’s exact test was used to assess the association between complication grades and the treatment group. The proportion of patients to have undergone a hepatic resection in each group was compared using a chi-squared test. Time-to-event between groups was compared using Kaplan-Meier curves, Cox proportional hazards models (the proportional hazards assumption was tested using Schoenfeld residuals) and log-rank tests. Median follow-up for the entire study period among those resected was estimated using the reverse Kaplan-Meier method. A Mann-Whitney U test was used to compare tumour characteristics between groups, such as the percentages of viable tumour in SIRT group vs. Chemotherapy only group. A two-sided 5% significance level was used for all tests. Statistical analyses were performed in STATA 15 and GraphPad Prism version 6.0 (GraphPad Software, La Jolla, CA, USA). 

## 5. Conclusions

We report the safety of hepatic resection after the novel combination of systemic OxMdG chemotherapy (oxaliplatin, 5-fluorouracil and folic acid) with SIRT in the first-line management of liver-dominant metastatic colorectal cancer, compared to OxMdG chemotherapy alone, and we present an exploratory analysis of histopathology. A Multi-Disciplinary Team deemed all patients inoperable before registration. FOXFIRE randomised 182 participants to chemotherapy alone and 182 to chemotherapy with SIRT. There was no statistically significant difference between groups in the rate of liver surgery, nor in survival from time of resection (HR = 1.55; 95% CI = 0.83–2.89). Ectopic microsphere deposition in the gall bladder was an incidental histological finding. Microsphere density was highest at the tumour periphery. Patients treated with SIRT plus chemotherapy displayed lower values of viable tumour in comparison to those treated with chemotherapy alone (*p* < 0.05). We conclude that liver surgery should be considered following the clinical response to SIRT. 

## Figures and Tables

**Figure 1 cancers-11-01155-f001:**
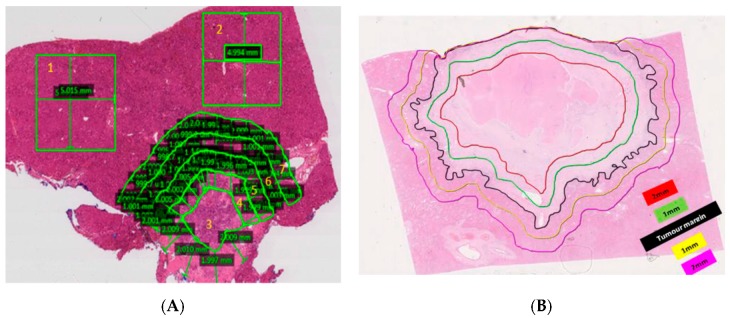
Example of zonal analysis on histopathology. (**A**) Method that was developed for calculation of microsphere deposition density. 1 and 2 = non-neoplastic tissue (NNT), 3 = Tumour centre, 4 = 1–2 mm zone within tumour border, 5 = 0–1 mm zone within tumour border, 6 = 0–1 mm zone away from tumour border, 7 = 1–2 mm zone away from tumour border. Slide magnification = 0.3×. (**B**) Illustration of the 1 mm and 2 mm zones from the tumour margin that microsphere densities were measured within. The highest density was found within 1 mm from the tumour periphery.

**Figure 2 cancers-11-01155-f002:**
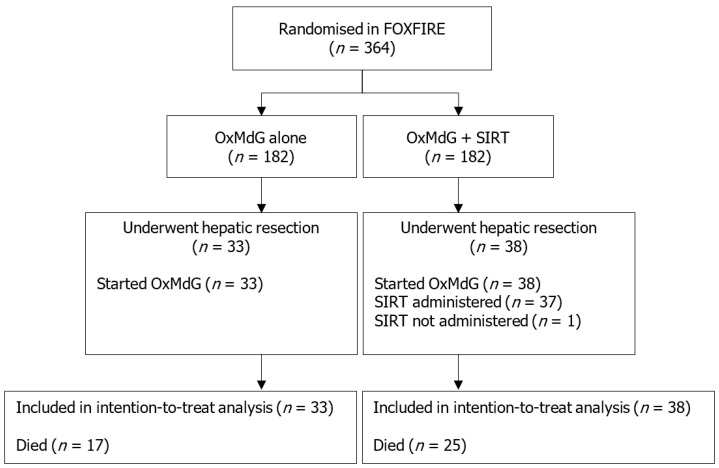
Study flow chart. OxMdG: Oxaliplatin, 5-fluorouracil and folic acid. SIRT: Selective internal radiotherapy. One patient in the OxMdG + SIRT group who underwent hepatic resection had started OxMdG chemotherapy, but did not receive SIRT.

**Figure 3 cancers-11-01155-f003:**
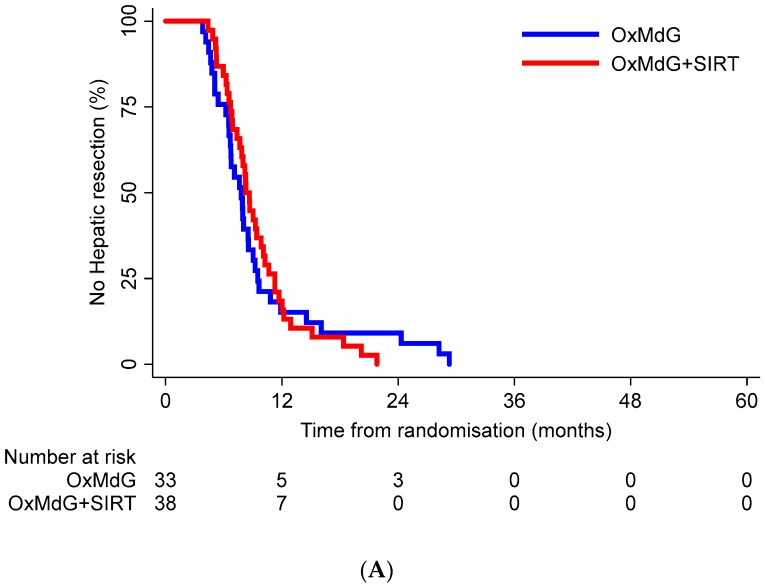
Kaplan-Meier curves for time to resection (**A**) and time to death (**B**). (**A**) Kaplan-Meier curve of time from randomisation to hepatic resection for each treatment group during the FOXFIRE study duration. (**B**) Kaplan-Meier curve of time from hepatic resection to death for each treatment group during the FOXFIRE study duration.

**Figure 4 cancers-11-01155-f004:**
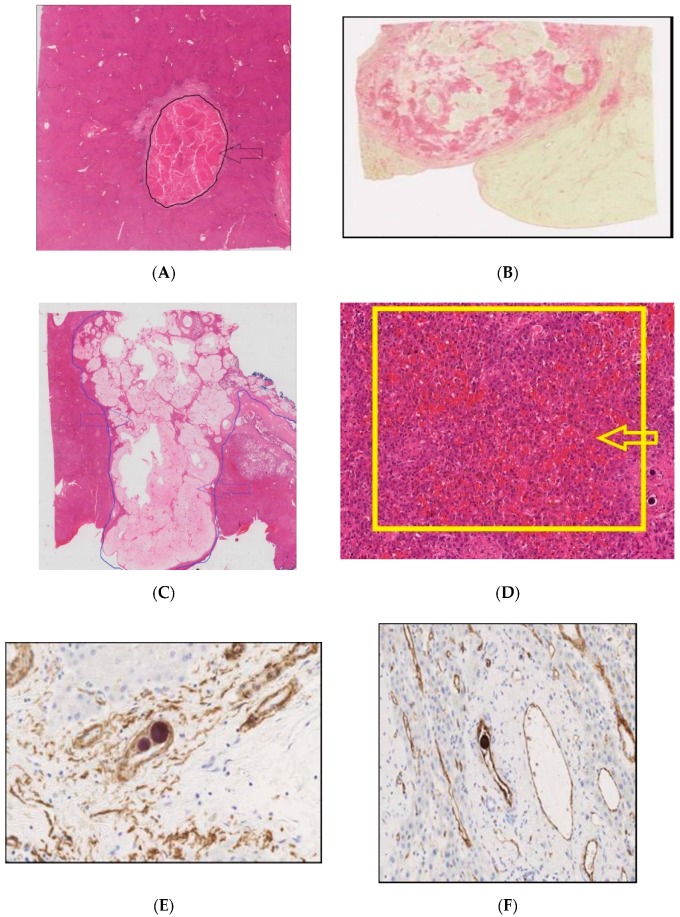
Histological studies of tumour and non-tumour tissues from patients receiving chemotherapy plus SIRT and proceeding to hepatic resection. (**A**) Complete tumour necrosis. Necrotic tissue enclosed in the black arrowed and delineated area (Haematoxylin and Eosin stain, original magnification ×0.7). (**B**) Extensive tumour fibrosis within tumour; collagen stained red (Sirius Red stain). Original magnification ×100. (**C**) Widespread mucinous change. Mucinous tissue enclosed in the blue arrowed and delineated area (Haematoxylin and Eosin stain, original magnification ×0.5). (**D**) Non-neoplastic liver tissue displaying vascular changes with marked sinusoidal dilatation and haemorrhage. Severe vascular changes enclosed in yellow-arrowed, delineated area at original magnification ×10. (**E**) Microspheres present in arterioles (identified by the smooth muscle in their walls using immunohistochemical staining for smooth muscle actin). Original magnification ×200. (**F**,**G**) Immunohistochemical staining for cells using endothelial CD31 stain. Original magnification ×200. (**H**) Section stained for the macrophage marker, CD68, to highlight the macrophage reaction to the microspheres. Original magnification ×400.

**Table 1 cancers-11-01155-t001:** Demographics of the study participants who had hepatic resection during the FOXFIRE clinical trial.

Characteristic at Baseline ^1^		OxMdG	OxMdG + SIRT	Total
		(*n* = 33)	(*n* = 38)	(*n* = 71)
Age at randomisation (years)		62 (42–78)	61 (30–83)	61 (30–83)
Extra-hepatic metastases status	No	28 (85%)	31 (82%)	59 (83%)
	Yes	5 (15%)	7 (18%)	12 (17%)
Degree of liver involvement	≤25%	27 (82%)	29 (76%)	56 (79%)
	>25%	6 (18%)	9 (24%)	15 (21%)
Gender	Male	22 (67%)	25 (66%)	47 (66%)
	Female	11 (33%)	13 (34%)	24 (34%)
Ethnicity	White or Caucasian	32 (97%)	33 (89%)	65 (93%)
	Asian	0 (0%)	2 (5%)	2 (3%)
	Black or African American	0 (0%)	1 (3%)	1 (1%)
	Other	1 (3%)	1 (3%)	2 (3%)
BMI (kg/m^2^)	≤30	23 (70%)	33 (87%)	56 (79%)
	>30	10 (30%)	5 (13%)	15 (21%)
WHO performance status	0	29 (88%)	24 (65%)	53 (76%)
	1	4 (12%)	13 (35%)	17 (24%)
Metastases present at initial diagnosis	No-Metachronous	8 (24%)	5 (14%)	13 (19%)
	Yes-Synchronous	25 (76%)	32 (86%)	57 (81%)
Primary tumour site	Left-sided primary	25 (86%)	31 (82%)	56 (84%)
	Right-sided primary	4 (14%)	7 (18%)	11 (16%)
Primary tumour in situ	No	14 (42%)	16 (42%)	30 (42%)
	Yes	19 (58%)	22 (58%)	41 (58%)
KRAS	Unknown	12 (36%)	10 (26%)	22 (31%)
	Wild Type	15 (45%)	23 (61%)	38 (54%)
	Mutation	6 (18%)	5 (13%)	11 (15%)
Prior adjuvant chemotherapy	No	31 (94%)	37 (97%)	68 (96%)
	Yes	2 (6%)	1 (3%)	3 (4%)
Prior pelvic radiotherapy	No	32 (97%)	36 (95%)	68 (96%)
	Yes	1 (3%)	2 (5%)	3 (4%)

^1^*n* (%) for categorical variables; median (min–max) for continuous variables. OxMdG: Oxaliplatin, 5-fluorouracil and folic acid; BMI: body mass index; WHO: World Health Organisation performance status; KRAS: Kirsten-RAS gene status.

**Table 2 cancers-11-01155-t002:** Classification of liver surgery performed during the FOXFIRE clinical trial. Surgical classification based on the Brisbane 2000 classification [12].

Type of Surgery Performed	Number of Surgical Episodes (Proportion of Total 82 Surgical Episodes)	Number of Surgical Episodes in Chemotherapy Alone Group	Number of Surgical Episodes in SIRT + Chemotherapy Group	Additional Procedures/Surgical Comments
Right hepatectomy	22 (27%)	9	13	Two patients had subsegmentectomies (segments IVa, V; II, III)
Left hepatectomy	9 (11%)	7	2	One patient had resection of inferior vena cava; One patient had subsegmentectomies (ns)
Segmentectomies	12 (15%)	7	5	Segmentectomies: Two patients: II, III; One patient each: III, VI, VII; III, IV, VII; IVb, V, VI; IV, V, VI (and subsegmentectomy and ablation); I, II, III; IV and ablation; IVa, VIII. One patient each with one segmentectomy: Vi; IV; II (subsegmentectomy III).
Extended right hemihepatectomy	7 (9%)	3	4	Three with subsegmentectomies (I; III; ns)
Extended left hemihepatectomy	1 (1%)	0	1	
Left lateral hepatectomy	2 (2%)	0	2	Atypical
Subsegmentectomies	9 (11%)	5	4	Three patients: ns. Two patients: Segment II only. One patient each: III; IVa, IVb, II; V, VI, VII (ablation II, IV)
No hepatic surgery report available	14 (17%)	5	9	Cholecystectomy in some cases
Open and closed	4 (5%)	2	2	
Other	2 (2%)	1	1	First part ALPPS. Right post sectionectomy (VI, VII).

Abbreviations used: Ns: Not specified; ALPPS: Associated liver partition and portal vein ligation for staged hepatectomy.

**Table 3 cancers-11-01155-t003:** Grading of surgical complications recorded in the FOXFIRE clinical trial. The table shows the grade for all patients who had complications post-hepatic resection using a classification developed for this indication [13]. Comparison of grades of complications between groups by Fisher’s exact test shows no significant difference between treatment groups (*p* = 0.380).

Grade	Number of Patients in Chemotherapy Alone Group	Number of Patients in Chemotherapy Plus SIRT Group
I	4	5
II	7	4
III	1	3
IVa	0	1
IVb	0	1
V	0	0

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
