# Peer review of "Hepatic Resection Following Selective Internal Radiation Therapy for Colorectal Cancer Metastases in the FOXFIRE Clinical Trial: Clinical Outcomes and Distribution of Microspheres"

_cancers, 2019, doi:10.3390/cancers11081155_

Round 1

Reviewer 1 Report

The authors present a retrospective analysis of patients undergoing hepatic resection in the randomized FOXFIRE trial, focusing on resection rates in both treatment arms, surgical morbidity, and histopathology findings in a subset of the whole surgical cohort (27 patients who gave consent). Although this is the first analysis of surgical aspects from patients treated in a randomized SIRT+chemo vs. chemo only trial, the feasibility of hepatic resection following SIRT has been demonstrated in several earlier reports and a retrospective analysis of surgical morbidity in a large cohort of patients resected after SIRT has been reported in the P4S study (cited by the authors). Hence, these results add to the existing body of evidence but the conclusion that “surgery after chemotherapy combined with SIRT is feasible" is not really new, and the relatively small numbers do not permit a valid comparison of surgical morbidity between the treatment arms. As the FOXFIRE study did not specifically enroll patients with a perspective for future hepatic resection (which explains the low rate of resections performed in both arms in this study and in the SIRFLOX trial), a formal comparison of resection rates between the treatment arms does not make sense, since many patient-, disease-, and surgeon-related  factors unrelated to treatment efficacy may have influenced the decision to resect or not to resect in an individual patient. By contrast, the analysis of histopathology findings in a subset of patients is interesting, as there are very few reports on this, and the paper should be refocused on these aspects if it is considered for publication.

Specifically, the following points should be addressed:

- Page 2, line 50: "Perioperative chemotherapy is considered the standard of care in potentially resectable CRCLM“. This is not exactly true: While PREoperative chemotherapy is the standard of care in unresectable CRCLM with a potential to become resectable after downsizing, the role of POSToperative chemotherapy after resection of CRCLM is controversial, as no single randomized trial has ever demonstrated its benefit. The reference cited by the authors is the EORTC40983 trial, which was a trial on perioperative chemotherapy in resectable (not potentially resectable) CRCLM and did not demonstrate a survival benefit in the ITT population. This is not an adequate reference to support any statement with regard to potentially resectable CRCLM that needs downsizing to become resectable.

- Page 2, line 69: „In this report, we examine whether the combination treatment improved resection rates for hepatic metastases…" These results were already presented in the original FOXFIRE publication (cited by the authors) and cannot be the main subject of the present publication. As stated above, FOXFIRE was not focused on candidates for hepatic resection, and a comparison of resection rates between treatment arms is therefore not meaningful.

- Figure 3: In Fig. 3A, the Y axis is marked “Hepatic Resection”; however, the figure shows two curves descending from 1.00 towards zero. This would mean that at 0 months, all patients had had hepatic resection, whereas at approx.. 30 months none had been resected. Certainly this is not what was intended by the authors and the Y axis legend should be modified accordingly. Moreover, in both Fig. 3A and 3B, the numbers on the Y axis (0.00 to 1.00) are not percentages (as implied by the Y axis legend in both graphs) but rather fractions of 1.

- Page 5, line 166: To avoid any misleading implications from the 0% and 50% medians of viable tumor from resection specimens in both treatment arms, these medians should be given with their ranges. Also, the statement in the abstract referring to this result (“Patients treated with SIRT plus chemotherapy displayed a 50% lower median value of viable tumor”) should not be left like this, as it implies that patients in the SIRT arm had half as much viable tumor as  patients in the chemo arm (medians should not be treated as and compared like continuous data).

- Page 8/9, line 206: “The results from the surgical outcomes…evaluation of patients receiving hepatic resection from both arms of the FOXFIRE trial have demonstrated that the rate of surgical complications and safety following the addition of SIRT is comparable to chemotherapy alone.” This statement is not supported by the results. There is no formal comparison of surgical complications in both treatment arms except for the list in table 3; moreover, for surgical morbidity to be comparable between treatment arms, the types of surgery performed in both arms must also be known. Types of surgery for the whole cohort but not for each arm separately are given in table 2.

Author Response

Specifically, the following points should be addressed:

- Page 2, line 50: "Perioperative chemotherapy is considered the standard of care in potentially resectable CRCLM“. This is not exactly true: While PREoperative chemotherapy is the standard of care in unresectable CRCLM with a potential to become resectable after downsizing, the role of POSToperative chemotherapy after resection of CRCLM is controversial, as no single randomized trial has ever demonstrated its benefit. The reference cited by the authors is the EORTC40983 trial, which was a trial on perioperative chemotherapy in resectable (not potentially resectable) CRCLM and did not demonstrate a survival benefit in the ITT population. This is not an adequate reference to support any statement with regard to potentially resectable CRCLM that needs downsizing to become resectable.

we have changed reference 3, as requested by the reviewer.

- Page 2, line 69: „In this report, we examine whether the combination treatment improved resection rates for hepatic metastases…" These results were already presented in the original FOXFIRE publication (cited by the authors) and cannot be the main subject of the present publication. As stated above, FOXFIRE was not focused on candidates for hepatic resection, and a comparison of resection rates between treatment arms is therefore not meaningful.

We have changed the emphasis of this statement, as requested by the reviewer.

- Figure 3: In Fig. 3A, the Y axis is marked “Hepatic Resection”; however, the figure shows two curves descending from 1.00 towards zero. This would mean that at 0 months, all patients had had hepatic resection, whereas at approx.. 30 months none had been resected. Certainly this is not what was intended by the authors and the Y axis legend should be modified accordingly. Moreover, in both Fig. 3A and 3B, the numbers on the Y axis (0.00 to 1.00) are not percentages (as implied by the Y axis legend in both graphs) but rather fractions of 1.

We have updated figure 3 as requested by the reviewer – a new version is attached to the resubmission.

- Page 5, line 166: To avoid any misleading implications from the 0% and 50% medians of viable tumor from resection specimens in both treatment arms, these medians should be given with their ranges. Also, the statement in the abstract referring to this result (“Patients treated with SIRT plus chemotherapy displayed a 50% lower median value of viable tumor”) should not be left like this, as it implies that patients in the SIRT arm had half as much viable tumor as  patients in the chemo arm (medians should not be treated as and compared like continuous data).

We have added the ranges to the main text, as requested by the reviewer, and have changed the sentence in the abstract.  the full data are already presented in supplementary table s2.

- Page 8/9, line 206: “The results from the surgical outcomes…evaluation of patients receiving hepatic resection from both arms of the FOXFIRE trial have demonstrated that the rate of surgical complications and safety following the addition of SIRT is comparable to chemotherapy alone.” This statement is not supported by the results. There is no formal comparison of surgical complications in both treatment arms except for the list in table 3; moreover, for surgical morbidity to be comparable between treatment arms, the types of surgery performed in both arms must also be known. Types of surgery for the whole cohort but not for each arm separately are given in table 2.

In order to clarify that a formal comparison was made, a statement has been added to the methods section and a sentence added to the table 3 legend.

As requested by the reviewer, table 2 has been modified to include the types of surgical episodes for each treatment arm separately. 

Reviewer 2 Report

The article entitled  ‘Hepatic resection following selective internal radiation therapy for colorectal cancer metastases in the FOXFIRE clinical trial: clinical outcomes and distribution of microspheres’ by Winter H et al describes the clinicopathological differences in patients receiving hepatic resection after combined SIRT+chemotherapy for colorectal metastases.

Though a very interesting concept, this study is limited due the high patient variability and relative low number of patients included.  

·       There are 82 surgical episodes among the 71 resected patients. Not clear which tissue samples were included in the histopathological assessment. Why is there one patient included in the chemo+SIRT group but did not receive SIRT (Figure 2)? What is the effect of bevacizumab and cetuximab? Some patients already had liver complications. Different kinds of surgery. How does this affect the selected 27 patients included for the histopathologic analysis? Any significant differences between the two groups? An additional table with patient characteristics for the histology group with p-values should be included. If possible, subgroups of the 27 patients should be analysed.

·       Not clear how the area of viable tumour is quantified. Only based on morphology? Representative images should be added and detailed description should be added to the M&M.

·       Examples of slides quantified for microspheres using the Aperio system should be included in the supplementary file. (overview and zoomed micrographs) How do you pick up microspheres? Manually or software based? Is there a threshold? Is one single particle regarded as positivity in an area? This part needs to be clarified.

·       Fig4C: You cannot claim there is a ‘mucinous change’ merely based on an H&E stain. An additional stain would be in place (e.g. PAS).

·       Fig4H: Image of the CD68 shows a lot of background and is of low resolution, hence not convincing at this stage. New images/stainings are in place.   

·       CD31/aSMA/CD68 staining should be quantified and correlated with percentage of viable tissue and number of microspheres.

Minor

·       P-values should be added to table 1 or results should be mentioned in the main manuscript.

Overall a good study. The variability in patient samples and some applied histological quantifications need to be clarified, and the immunohistochemistry requires further analysis.  

Author Response

There are 82 surgical episodes among the 71 resected patients. Not clear which tissue samples were included in the histopathological assessment. Why is there one patient included in the chemo+SIRT group but did not receive SIRT (Figure 2)? What is the effect of bevacizumab and cetuximab? Some patients already had liver complications. Different kinds of surgery. How does this affect the selected 27 patients included for the histopathologic analysis? Any significant differences between the two groups? If possible, subgroups of the 27 patients should be analysed.

Figure 2 shows the intention-to-treat analysis, as is mandatory for consort diagrams and for presentation of data from randomised controlled trials.

Table 2 has been modified to include the types of surgical episodes for each treatment arm separately

Please note that the total number of patients proceeding to surgery in each treatment group is smaller than the number of surgical episodes since some patients had surgical episodes more than once.

Unfortunately, the sample size is too small to study subgroups.

·       Not clear how the area of viable tumour is quantified. Only based on morphology? Representative images should be added and detailed description should be added to the M&M.

The reviewer correctly states that is has been done based on morphology.  This is the standard method for histopathology.  There is no standard stain for dead tissue.

Representative images should be added and detailed description should be added to the M&M.

Examples of slides quantified for microspheres using the Aperio system should be included in the supplementary file. (overview and zoomed micrographs) 

As requested by the reviewer, a new supplementary figure s1 has been added.

How do you pick up microspheres? Manually or software based?

Microspheres were counted manually – this has been added to the methods section.

•       Fig4C: You cannot claim there is a ‘mucinous change’ merely based on an H&E stain. An additional stain would be in place (e.g. PAS).

An alternative figure 4c has been prepared and is attached

•       Fig4H: Image of the CD68 shows a lot of background and is of low resolution, hence not convincing at this stage. New images/stainings are in place.  

A new image for figure 4h has been prepared and is attached

•       CD31/aSMA/CD68 staining should be quantified and correlated with percentage of viable tissue and number of microspheres.

This could not be performed since different areas were analysed for the respective stains 

Minor

·       P-values should be added to table 1 or results should be mentioned in the main manuscript.

Table 1 is a baseline characteristics table showing the balance/difference (in patient characteristics) between randomised groups at baseline, i.e. before patients start any treatment - you would not expect to see any differences before treatment is started so having p-values here does not make sense. 

According to the study statistician, p-values are not relevant analyses for such a small study exploring the effect of adding sirt to chemotherapy in the context of hepatic resections.  Unfortunately, the sample size is too small to study the effect of baseline factors on outcomes. Assessing the effect of patient characteristics on outcomes is more appropriate in a larger sample size than the one available for this study. 

Overall a good study. The variability in patient samples and some applied histological quantifications need to be clarified, and the immunohistochemistry requires further analysis. 

Thank you to the reviewer for the positive feedback and suggestions to improve the manuscript prior to publication.

Round 2

Reviewer 1 Report

The authors have addressed most of the points raised during the review of the original manuscript. I still believe that the main interest of the paper lies in the histopathology substudy rather than the surgical outcome results, as the feasibility of hepatic resection following SIRT is not really a new finding. I'd also recommend rephrasing the statement that perioperative chemotherapy is considered the standard of care in potentially resectable CRCLM, as this is not unequivocally the case. 

Author Response

The authors have addressed most of the points raised during the review of the original manuscript. I still believe that the main interest of the paper lies in the histopathology substudy rather than the surgical outcome results, as the feasibility of hepatic resection following SIRT is not really a new finding. I'd also recommend rephrasing the statement that perioperative chemotherapy is considered the standard of care in potentially resectable CRCLM, as this is not unequivocally the case.

Responses:

We have changed the sentence referring to peri-operative chemotherapy, as requested by the reviewer. 

We agree with the reviewer that the histopathology substudy is an important feature of this paper, which is why we have discussed these data in detail and we have emphasised this point in the last paragraph of the dicussion section as follows:

“This report represents the most detailed description of surgical and histopathological findings of patients treated prospectively within a clinical trial of chemotherapy plus SIRT for CRCLM.”

The study is CONSORT-compliant and a CONSORT diagram is included as one of the figures.  Trial registration number and ethics approval number are stated in the manuscript.  The study was funded Cancer Research UK and an unrestricted grant from Sirtex Medical; and the design, conduct and analysis has been entirely independent of manufacturer influence. The study was sponsored by the University of Oxford.  Competing financial interests and conflicts have been described in full in the manuscript.  We acknowledge that all authors contributed to this paper sufficiently to be named as authors, according to international guidance. The results have not been published in another journal.